# Analysis of Halophilic Phenotypic Variation and Cytotoxicity of *Vibrio parahaemolyticus* from Different Sources

**DOI:** 10.3390/pathogens14020182

**Published:** 2025-02-12

**Authors:** Jingyue Gu, Xin Dong, Yunqian Zhou, Ying Zhao, Qiang Du, Jia Chen, Xujian Mao, Fengming Wang, Bowen Tu

**Affiliations:** 1School of Public Health, Xuzhou Medical University, Xuzhou 221004, China; 302110120948@stu.xzhmu.edu.cn; 2Changzhou Centre for Disease Control and Prevention, Changzhou Medical Center, Nanjing Medical University, Changzhou 213000, China; dongxin770099@163.com (X.D.); my19880729@163.com (Y.Z.); bodezhilang@sina.com (Q.D.); 18851400177@163.com (J.C.); m5575383@163.com (X.M.); wfm0519@163.com (F.W.); 3School of Public Health, Nanjing Medical University, Nanjing 211166, China; 2024120921@stu.njmu.edu.cn

**Keywords:** foodborne disease, freshwater products, aquatic animal pathogen evolution, host-pathogen interaction, virulence alteration, low-saline water

## Abstract

*Vibrio parahaemolyticus* is an aquatic animal pathogen. Recently, the detection rate of *V. parahaemolyticus* in freshwater products has exceeded that in seafood products, and the strains isolated from freshwater products exhibit better growth conditions in low-salinity environments. This study is based on a food risk detection activity in Changzhou, Jiangsu Province, China, investigating the variation of halophilism and the virulence of two groups of strains under different salt concentrations. Under 0%, 0.5%, and 1% salt, the strains from the freshwater showed faster growth than those from the seawater. In comparison, the strains from the seawater group under 2% and 3% salt grew faster than the growing status under the foregoing low-salt concentration environment. The cytotoxicity produced by the two strains was approximately 1.4 times higher in the 0.5% and 1% salt concentration groups compared to the 3% corresponding experimental group. Under the 0%, 0.5%, and 1% salt, the cytotoxicity of strains in the freshwater group increased by nearly 20% compared to that in the seawater groups. The freshwater strains showed altered halophilism and adapted to the low-salt environment. This research will be helpful in establishing a local and global control strategy against the diseases resulting from *V. parahaemolyticus*.

## 1. Introduction

*Vibrio parahaemolyticus* is a significant globally concerned foodborne pathogen and is widespread in the marine environment and seafood worldwide [1,2,3,4]. Consumption of food contaminated with *V. parahaemolyticus* is highly susceptible to food poisoning [5,6], and the clinical manifestations of infection are usually nausea, vomiting, diarrhea, etc., and are even life-threatening [7]. The foodborne disease outbreak monitoring system in China showed that the number of cases of *V. parahaemolyticus* infection exceeded 27,212 between 2010 and 2020 [8]. In the last three years, the number of foodborne diseases infected by *V. parahaemolyticus* in coastal areas such as Guangzhou City and Zhejiang Province has accounted for more than 50% of the total infections [9,10,11,12,13]. It is suggested that foodborne diseases caused by *V. parahaemolyticus* have emerged as a significant public health concern in coastal populations.

According to traditional bacteriology, *V. parahaemolyticus* is classified as moderately halophilic, with an optimal salt concentration for growth of approximately 3%. In comparison, halophilic experiments usually show growth inhibition at salt concentrations below 0.5% or above 8.5% [14,15]. Consequently, the historical data in Chinese Food Risk Monitoring data show a high level of *V. parahaemolyticus* contamination in seafood [16]. With the continuous development of China’s aquaculture economy, industrial habits, such as the landing of aquaculture and sharing of aquaculture water for fresh and marine products, have led to the continuous migration of *V. parahaemolyticus* growth environments from seawater towards freshwater environments [17]. This study is based on the results of the Changzhou Food Risk Monitoring project from 2015 to 2022. The foodborne pathogen detection rate of *V. parahaemolyticus* increased from 2.5% to 43%. The results showed that the detection rate of *V. parahaemolyticus* in freshwater products exceeded that of seafood, accounting for 40% of all samples. The detection rate of the abnormal rise of *V. parahaemolyticus* isolated from freshwater products was also reported. It was indicated that freshwater products have been seriously contaminated with *V. parahaemolyticus* in recent years [18,19,20,21]. It is hypothesized that *V. parahaemolyticus* may continue to tolerate and change in low-salt environments, and their virulence and other metabolisms were gradually changing while the strains adapted to freshwater habitats. In this study, *V. parahaemolyticus* detected in the food risk monitoring in Changzhou between 2015 and 2022 were classified and cultured based on their sources, and their biological characteristics and cytotoxicity differences were preliminarily explored to warn of their significance for monitoring in the field of public health.

*V. parahaemolyticus* is a pathogen that infects human intestinal epithelial cells. Caco-2 cells show characteristic intestinal epithelial cell differentiation when full-grown, with a microvillous structure and typical epithelial cell morphology, and share similar structural and functional properties with human small intestinal epithelial cells, which are commonly used to assess the toxic effects of bacteria on intestinal cells. Therefore, the Caco-2 cell line was used to assess the enterotoxicity and cytotoxicity of *V. parahaemolyticus*. Commonly used methods to detect Vibrio cytotoxicity include MTT, LDH, and CCK8. CCK8 and LDH methods are more sensitive, rapid, and stable, and LDH and CCK8 methods were chosen for *V. parahaemolyticus* cytotoxicity detection in this study [22]. Simultaneously, this study compares the virulence differences between freshwater and seawater strains using the previously established halophilic unaltered *V. parahaemolyticus* standard strains from the literature. Additionally, a systematic analysis of the growth patterns and cytotoxicity of the strains across various salinity conditions was performed through a range of salt concentrations.

## 2. Materials and Methods

### 2.1. Strains and Cell Sources

The strains were obtained from the isolates identified and characterized in the food safety monitoring initiative conducted in Changzhou from 2015 to 2022 (Table 1), including freshwater and seafood sources. Caco-2 (BeiNaBio, Beijing, China, BNCC350769), Hela (BeiNaBio, Beijing, China, BNCC342189), and the *V. parahaemolyticus* standard strain (BeiNaBio, Beijing, China, ATCC17802) were used in the following experiments.

### 2.2. Strain Culture and Identification

The food risk monitoring samples, such as freshwater fish, deep-sea yellowtail, etc. were collected from five districts of Changzhou. Strains were isolated and cultured in accordance with the National Standard for Food Safety of Food Microbiology Inspection for *V. parahaemolyticus* (GB4789.7-2013).

Under aseptic conditions, the samples were added to the enrichment solution and homogenized, and then the strains were placed at 37 °C in an incubator for 18 h. The next day, the samples were inoculated with TCBS medium plate (Hopebio, Qingdao, China, HBPM9301) and Vibrio chromogenic medium plate (Hopebio, Qingdao, China, HB0109-25) by using an inoculating ring under the surface of the enrichment solution. The mediums were then placed in an incubator at 37 °C for 24 h. *V. parahaemolyticus* showed round, smooth-surface green colonies on TCBS. Freshwater and seawater strains identified as *V. parahaemolyticus* were classified and stored in an ultra-low-temperature refrigerator at −80 °C by glycerol preservation method ready for subsequent experimental studies.

### 2.3. Determination of Growth Curves

A single colony of every strain was picked and resuspended into the Luria–Bertani liquid medium (LB) (Beijing Bio, Beijing, China) containing 0%, 0.5%, 1%, 2%, 3%, and 5% sodium chloride, and then standardized configuration to a bacterium suspension with 0.5 McFarland (McF) (pH: 7.0). After thorough mixing, 200 μL of the suspension was transferred to 96-well plates (BeyoGold, Shanghai, China, FCP962), which were aseptically laminated and incubated at 37 °C for 24 h [23]. The absorbance of each well was measured at 600 nm [24] (Tecan, Männedorf, Switzerland, Infinite 200pro) at hourly intervals with parallel detection three times. We measured the data once at each time point for the two groups of strains and repeated it three times in parallel. The data from each time point were averaged as the absorbance value. We fit a nonlinear regression curve with the cultivation time as the horizontal axis and the absorbance value of the culture as the vertical axis. We drew the halophilic growth curves of *V. parahaemolyticus* at different concentrations mentioned above.

### 2.4. Establishment of Caco-2 and Hela Cell Infection Models

Caco-2 and Hela cells were inoculated into 6-well plates equipped with transwell chambers (Corning, New York, NY, USA, 3428) at a density of 9 × 10^4^ cells/well; a volume of 100 μL of cell suspension was added to the upper layer of the chambers, and 500 μL of Dulbecco’s Modified Eagle Medium (DMEM) (Gibco, Thermo Fisher Scientific, Waltham, MA, USA), supplemented with 10% Fetal Bovine Serum (FBS) (Gibco, A5256701), were added to the lower layer of the chambers, and incubated for 24 h at 37 °C with 5% CO_2_ (Thermo, Waltham, MA, USA, Forma Direct Heat CO_2_ Incubators).

### 2.5. Standardized Cytotoxicity Assay

A certain amount of 3% sodium chloride LB medium was taken, and the standard strain of *V. parahaemolyticus* ATCC17802 was revived by activation culture. Positive strains identified as *V. parahaemolyticus* by the microbial mass spectrometry were inoculated into fresh LB medium and prepared as 0.5 McFarland (MCF) suspension, which was incubated at 37 °C with 180 r/min shaking for 5~6 h. Whereafter, cultures were centrifuged at 4500 r/min for 10 min, and the precipitates were resuspended in 10% FBS in DMEM medium to establish a cell infectious model at a multiplicity of 100:1 [25]. Lactate dehydrogenase (LDH) release from the cells was assayed at 2 h intervals to assess bacterial cytotoxicity (LDH cytotoxicity test kit, MCE, Monmouth Junction, NJ, USA, HY-K1090-500T; CCK8 kit, GLPBIO, Montclair, CA, USA, GK10001), and was monitored continuously up to 24 h. The time when LDH release significantly increases was selected for subsequent correlation analyses between salt concentration and cytotoxicity. The high control group represents the maximum releasable LDH after cell lysis, the high control blank group was deducted from the background absorbance of the high control group, the low control group represents the spontaneous release of LDH from untreated cells, and the background blank group was used to deduct the background absorbance values of the low control and the sample wells.

### 2.6. Correlation Analysis Between Salt Concentration and Cytotoxicity

Strain inoculation and cytotoxicity assays were conducted to evaluate the variations in cytotoxicity associated with different salt concentrations and the respective strains. The strains exhibiting the most pronounced growth within each category were selected as subjects for toxicity assessment (including freshwater strains VP1−12 and seawater strains VP13−14). They were inoculated into 0.5%, 1%, and 3% sodium chloride LB for overnight incubation, respectively.

### 2.7. Data and Statistical Analyses

Data from each experimental group were analyzed, and the curve was fitted using GraphPad prism 10.0. The data were processed using SPSS 27.0 for significant difference analysis. A two-sample independent *t*-test was used to analyze the differences in the growth curves of the two groups of strains. The cytotoxicity values were statistically tested by one-way ANOVA, multifactorial ANOVA, and Bonferroni two-by-two comparisons to analyze the statistical significance of the differences between the groups (*p* < 0.05 was significantly different, and *p* < 0.01 was highly significantly different).

## 3. Results

### 3.1. Detection of Vibrio parahaemolyticus in Changzhou from 2015 to 2022

The detection rate of *V. parahaemolyticus* increased from 2.5% to 43% during the 8 years. The prevalence of *V. parahaemolyticus* in freshwater products increased from 0% to 45% and in seawater products from 5% to 41% (Figure 1). The number of food poisoning cases caused by *V. parahaemolyticus* also increased from 0 to 6 times between 2015 and 2022 (Figure 2).

### 3.2. Differences in the Growth of Strains at Different Salt Concentrations

The freshwater strains exhibited a more rapid growth rate compared to the seafood group in 0%, 0.5%, and 1% sodium chloride LB (0%, *p* = 0.0102; 0.5%, *p* = 0.0077; 1%, *p* = 0.0490). Conversely, the seawater group strains demonstrated superior growth in 2% and 3% sodium chloride LB (2%, *p* = 0.0317; 3%, *p* = 0.0446). When two groups of strains grew in 5% sodium medium LB, there was a negligible significant difference in the bacterial densities between the two groups (5%, *p* = 0.8703 > 0.05) (Figure 3).

### 3.3. Cytotoxicity Standardization Results

The cytotoxic effects of the standard strain of *V. parahaemolyticus*, ATCC17802, were observed to escalate from 2 h post-infection and remained elevated until the 12 h mark (Figure 4). Caco-2 cells exhibited a greater degree of cytotoxicity following infection than Hela cells. After a 6 h exposure to the standard strain of *V. parahaemolyticus*, there was a notable increase in LDH release in both cell types. Consequently, the correlation between cytotoxicity and various treatments was assessed at the 6 h post-infection interval.

### 3.4. Correlation Between Salt Concentration and Cytotoxicity

The freshwater group strains exhibited the highest cytotoxicity at 0.5% sodium chloride solution, followed by 1%, and the lowest cytotoxicity at 3% under both Caco-2 and Hela cell models (*p* < 0.001, Appendix A) (Figure 5A). In the same cellular model, the seawater group strains also exhibited the strongest cytotoxicity at 0.5% and the lowest cytotoxicity at 3% (*p* < 0.001) (Figure 5B). The freshwater group strains showed more cytotoxic than the seawater group strains under the foregoing three salt concentrations (*p* < 0.001). The cell viability assay results further validated that at equivalent salt concentrations, the viability of strains in the freshwater cohort was significantly lower compared to that of the strains in the seawater cohort (Figure 5C).

## 4. Discussion

Halophilism is one of the most unique biological characteristics of *V. parahaemolyticus*, and the optimal salt concentration for its growth is 3%, leading to a higher detection rate in seafood over the past few decades. However, recent trends indicate an increased contamination of *V. parahaemolyticus* in freshwater products, surpassing its detection rate in seafood. Meanwhile, a series of studies found that the bacterial density of *V. parahaemolyticus* cultured at 0.66% and 2% sodium chloride solution was the same, and the growth rate of the bacterium at 0.5% was even faster than that at 3% [26,27,28]. In addition, the expression of osmotically adapted genes, such as *ompU* and *ompN*, were upregulated at 0.5% salt concentration [29,30,31,32], all of which suggest that *V. parahaemolyticus’* salt adaptation is altered to adapt to low-salt environments [33,34]. However, recent studies have primarily focused on the clinical isolates of *V. parahaemolyticus* [35], and the altered salt responsiveness of *V. parahaemolyticus* from other sources is poorly understood.

In this experiment, the halophilism characteristics of *V. parahaemolyticus* from different sources were investigated, in which the strains in the freshwater group grew faster than the strains in the seawater group at low sodium chloride concentration solutions (0%, 0.5%, and 1%), and when the salt dependence was varied. Previous studies only discussed the growth of the same *V. parahaemolyticus* strain in two sodium chloride solutions and found that the strain grew faster at low-salt concentrations than in the 3% group [36]. In the current investigation, the strain classifications and sources were meticulously refined to assess the growth dynamics of each strain from diverse origins across a spectrum of sodium chloride concentrations. It was observed that both strain groups maintained optimal growth at 3% salinity. However, the freshwater strains demonstrated greater adaptability to low-salinity environments compared to their seawater counterparts. This finding implies that *V. parahaemolyticus* may have undergone adaptations to freshwater ecosystems during its transition from marine to freshwater habitats, enabling its survival in freshwater environments and associated products.

The abnormal growth exhibited by *V. parahaemolyticus* in low-salt environments has led to further adaptation to low-salt environments and continuous invasion of the human life cycle on the one hand; on the other hand, the cytotoxicity produced by the bacterium in low-salt environments is further altered [37]. *V. parahaemolyticus* cultured at 0.5% or 1% sodium chloride solution has been reported to show higher cytotoxicity than that at 3% [24], although the current study only compared two sets of salt concentrations, 0.5% versus 3%, or 1% versus 3%. In the present experiment, two infection models, Caco-2 and Hela cell lines [27], were used as a framework to explore the differences in *V. parahaemolyticus* cytotoxic effects at the cellular level, and sodium chloride solutions of 0.5%, 1%, and 3% were included in the same experiment for a comprehensive comparison. Meanwhile, this experiment was conducted to analyze the difference in cytotoxicity between freshwater and seawater strains by comparing them with a standard strain of *V. parahaemolyticus*, whose traditional trait, i.e., halophilism, had not been changed. The results showed that both types of strains were most cytotoxic at 0.5%, followed by 1%, and it is possible that the low-salt environment induced increased cytotoxicity in the strains. The literature suggests that ToxR is an important transcriptional regulator of *V. parahaemolyticus*, modulating the expression of cytotoxicity in this bacterium. The unique feature of this experiment lies in the successful screening of mutant strains through a series of salt concentration settings, and the cytotoxic difference between freshwater and seawater strains at high and low salt concentrations was distinguished, compared to previous experiments that only used standard strains. However, the strains used in the experiment only originated from Changzhou and were not expanded to other regions, so there are some limitations. Nevertheless, the results of this experiment provide some theoretical references for further research on *Vibrio parahaemolyticus*. Based on the experimental results, it is tentatively hypothesized that the increase in cytotoxicity of the strain in a low-salt environment is regulated through ToxR [34]. In addition, compared with the strains in the seawater group, the strains in the freshwater group were more cytotoxic in all groups. There was an increased risk of food poisoning due to greater daily exposure to freshwater products.

## 5. Conclusions

In summary, *V. parahaemolyticus* in freshwater products exhibited enhanced adaptability and cytotoxicity at reduced salinity levels, with its halophilic traits undergoing modification during the transition from marine to freshwater habitats. While the molecular basis for this altered halophilism remains unclear, this study has elucidated the biological characteristics of both freshwater and seawater strains across varying salt concentrations, thereby providing a theoretical framework for future comprehensive investigations of this bacterium in both low and high-salinity environments. Furthermore, there is a pressing need to intensify the systematic surveillance of *V. parahaemolyticus* from diverse sources to effectively mitigate the contamination risks associated with this pathogen in freshwater products and safeguard public health.

## Figures and Tables

**Figure 1 pathogens-14-00182-f001:**
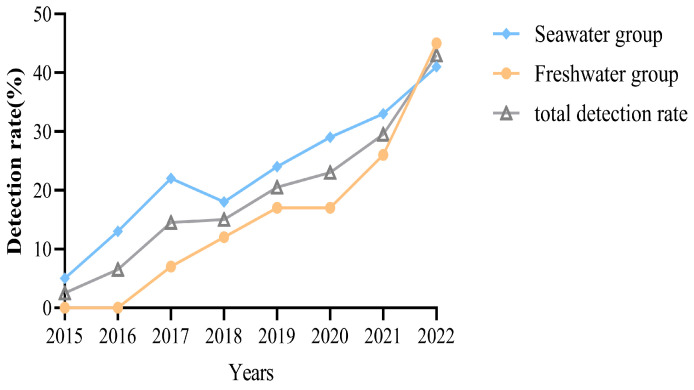
*Vibrio parahaemolyticus* detection rate in food risk monitoring in Changzhou from 2015 to 2022. Freshwater group strains (yellow), seawater group strains (blue), total detection rate (gray).

**Figure 2 pathogens-14-00182-f002:**
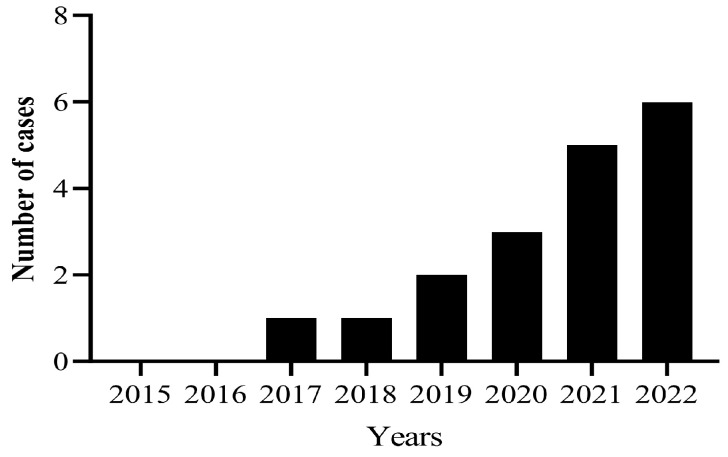
Number of food poisonings caused by Vibrio parahaemolyticus from 2015 to 2022.

**Figure 3 pathogens-14-00182-f003:**
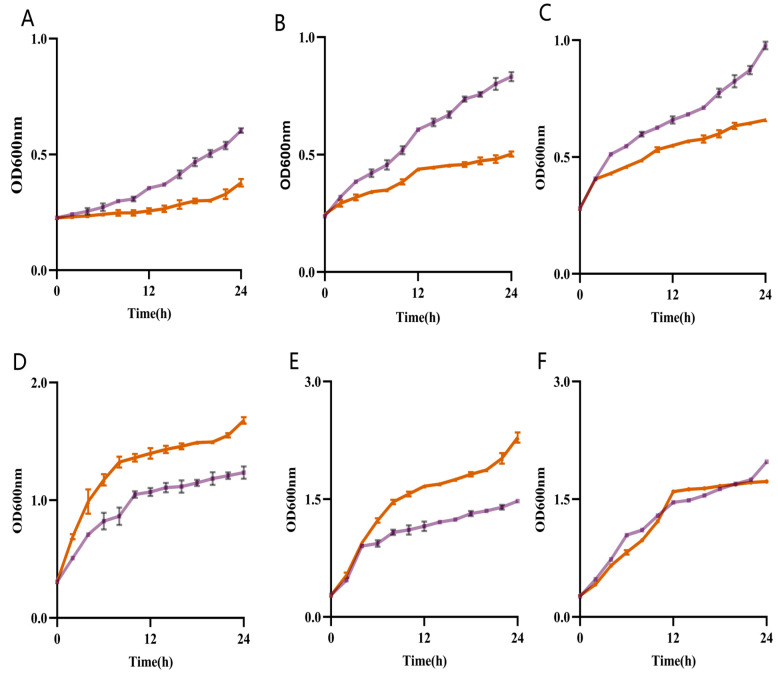
Growth curves of freshwater and seawater group strains cultured in 0% (**A**), 0.5% (**B**), 1% (**C**), 2% (**D**), 3% (**E**), and 5% (**F**) sodium chloride LB liquid medium for 24 h. Freshwater group strains (purple), seawater group strains (orange).

**Figure 4 pathogens-14-00182-f004:**
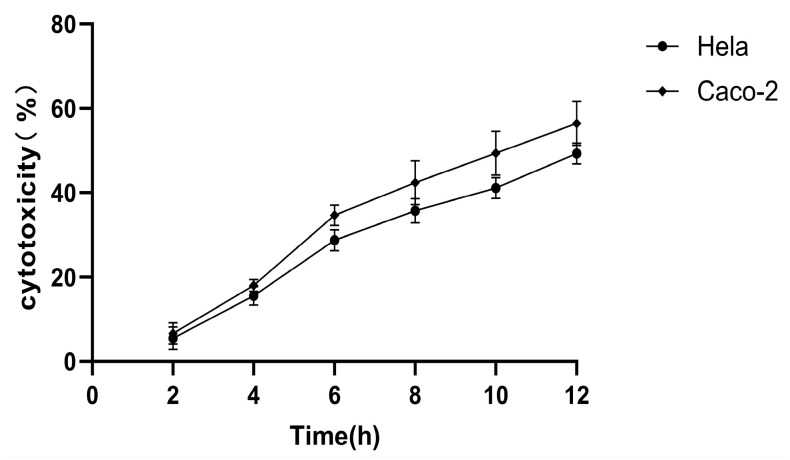
Cytotoxicity of *Vibrio parahaemolyticus* standard strain ATCC17802 after infection of Caco-2 and Hela cells. ● (Hela cells), ◆ (Caco-2 cells).

**Figure 5 pathogens-14-00182-f005:**
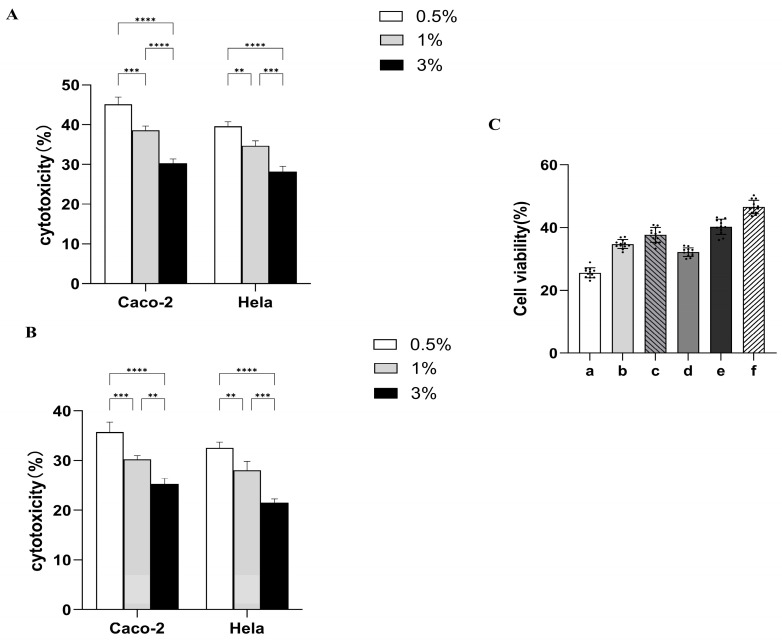
Cytotoxicity analysis of Vibrio parahaemolyticus strains cultured in 0.5%, 1%, and 3% sodium chloride LB liquid medium: (**A**) Freshwater group strains infected with Caco-2 and Hela at different sodium chloride solutions. (**B**) Seawater group strains infected with Caco-2 and Hela at three sodium chloride solutions. (**C**) Cell viability of both groups of strains after infection with Caco-2 at three sodium chloride solutions. a, b, c: freshwater strain: infected cells at 0.5%, 1%, and 3% salinity. d, e, f: seawater strain: infected cells at 0.5%, 1%, and 3% salinity. Data are expressed as mean ± standard deviation (Dots indicate sample size, *n* = 12). ** *p* < 0.01, *** *p* < 0.001, **** *p* < 0.0001.

**Table 1 pathogens-14-00182-t001:** Source and number of *Vibrio parahaemolyticus* strains used in the experiment.

STRAIN	SOURCE OF STRAIN	*TDH*/*TRH*/*TLH*	YEAR	REGION
VP1	River shrimp (freshwater product)	−/−/+	2022	Changzhou Economic Development Zone
VP2	Field snails (freshwater product)	+/−/+	2022	Changzhou Wujin
VP3	River shrimp (freshwater product)	−/+/+	2022	Changzhou Wujin
VP4	River shrimp (freshwater product)	−/−/+	2020	Changzhou Xinbei
VP5	Eels (freshwater product)	−/−/+	2022	Changzhou Xinbei
VP6	Mussels (freshwater product)	−/−/+	2016	Changzhou Economic Development Zone
VP7	River shrimp (freshwater product)	−/−/+	2016	Changzhou Zhonglou
VP8	Field snails (freshwater product)	−/−/+	2018	Changzhou Jintan
VP9	Mussels (freshwater product)	−/−/+	2015	Changzhou Tianning
VP10	Crucian carp (freshwater product)	−/−/+	2017	Changzhou Economic Development Zone
VP11	Crucian carp (freshwater product)	−/−/+	2018	Changzhou Economic Development Zone
VP12	Crucian carp (freshwater product)	−/−/+	2019	Changzhou Xinbei
VP13	Yellow croaker (seafood)	−/−/+	2020	Changzhou Jintan
VP14	Squid (seafood)	+/−/+	2019	Changzhou Zhonglou
VP15	Squid (seafood)	−/−/+	2019	Changzhou Economic Development Zone
VP16	Razor clams (seafood)	+/−/+	2022	Changzhou Economic Development Zone
VP17	Salmon (seafood)	−/−/+	2016	Changzhou Zhonglou
VP18	Salmon (seafood)	−/+/+	2018	Changzhou Jintan
VP19	Deep-sea silverfish (seafood)	−/−/+	2015	Changzhou Tianning
VP20	Large yellow croaker (seafood)	−/−/+	2022	Changzhou Zhonglou
VP21	Striped bass (seafood)	+/−/+	2022	Changzhou Economic Development Zone
VP22	Small yellow croaker (seafood)	−/−/+	2020	Changzhou Zhonglou
VP23	Tuna (seafood)	+/−/+	2022	Changzhou Jintan
VP24	Striped bass (seafood)	−/−/+	2016	Changzhou Economic Development Zone
VP (ATCC17802)	BeiNaBio	+/−/+		NA

## Data Availability

Data available on request from the authors.

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
