# Peer review of "Analysis of Halophilic Phenotypic Variation and Cytotoxicity of Vibrio parahaemolyticus from Different Sources"

_pathogens, 2025, doi:10.3390/pathogens14020182_

Round 1
Reviewer 1 Report
Comments and Suggestions for Authors
The study conducted by Gu et al. evaluated the growth and cytotoxicity of Vibrio parahaemolyticus strains from freshwater versus seafood samples. To improve the quality of the manuscript, I am sending below my recommendations for modifications.
Title: Put Vibrio parahaemolyticus in italics. Make this change throughout the manuscript.
Abstract:
Line 18: add "China", after "Province,"
Line 25: change "virulence" to "cytotoxicity"
Keywords: replace words that are already present in the manuscript title.
Introduction: Tell us a little more about the ways of evaluating cytotoxicity in isolates from aquatic animals, especially Vibrio spp., and why cell lines from humans are used in this type of evaluation and not cell lines from aquatic animals.
Material and Methods:
Lines 71-73: If the isolates used have been characterized in a previous study, please provide the appropriate citation. Otherwise, please explain how the samples were collected, how the bacteriological examination was performed and how they were identified at the species level.
Lines 78-79: add brand of culture media used.
Line 83: describe in full the first time LB appears in the manuscript.
Line 83-89: Was the experiment conducted in replicates? From Figure 1 it seems so, so I ask that you add the number of replicates tested.
Line 88: change "hole" to "well"; and was any analysis of colony forming unit counts performed on any of these ODs analyzed?
Line 98: Why didn't the authors perform the toxicity test with other concentrations (2 and 5%) of salt?
Lines 118-120: This sentence should come at the beginning of the paragraph and not at the end.
Line 123: with 10 versions or version 10 of the program?
Results:
Line 130: delete "status"
Figure 1: improve the quality of the figure; subtitle suggestion: "Growth curves of freshwater and seawater group strains cultured in 0% (A), 0.5% (B), 1% (C), 2% (D), 3% (E), and 5% (F) sodium chloride LB liquid medium for 24 hours. Freshwater group strains (purple), seawater group strains (orange)."
Line 158: change "Sodium" to "sodium". Make this change throughout the manuscript.
Figure 3: Why was only Caco-2 cell line chosen in image 3C?
Table 1: move to after item 2.1; change "Regin" to "Region"
Discussion:
Line 178: change "Haliphilism" to "Halophilism"; In addition, the word halophilism is repeated several times from the beginning of the discussion... would it be possible to change some of them for another synonym?
Line 185: if OmpU and OmpN are referencing genes and not synthesized proteins, they should be left in italics.
Line 214: change "virulence" to "cytotoxicity"
Line 221: What types of limitations?
The authors mentioned that the growth of Vibrio parahaemolyticus in different salt concentrations has already been evaluated... so, what is new about your study? The authors need to value the originality of the study carried out in the discussion.
Conclusions:
Line 230: change "virulence" to "cytotoxicity"
Author Response
Response to Reviewer
Thank you very much for your insightful comments and valuable suggestions. The suggestions make our study more comprehensive and more rigorous. Based on the feedback provided, we have thoroughly revised the manuscript, and all authors have approved the changes. The main changes in the text are highlighted in red.
We have detailed the point-by-point response to the comments and suggestions below.
Comments 1: [Title: Put V. parahaemolyticus in italics. Make this change throughout the manuscript.]
Response:We sincerely appreciate your careful reading! We have put the “V. parahaemolyticus” in italics throughout the manuscript.
Comments 2: [ Abstract: Line 18: add “China”, after “Province,”]
Response:Thanks for your careful check. We are sorry for our carelessness. Based on your suggestion, we have added “China” after “Jiangsu Province”.(Page 1, Abstract, Line 19)
Comments 3: [Abstract: Line 25: change “virulence” to “cytotoxicity”]
Response:Thank you for your careful check. We revised the “cytotoxicity” instead of “virulence”. (Page 1, Abstract, Line 26)
Comments 4: [Keywords: replace words that are already present in the manuscript title.]
Response:We would like to thank you for your advice. We have made careful modifications to the keywords, replacing “Vibrio parahaemolyticus” with “Foodborne disease” and “Cytotoxicity” with “Freshwater products” to ensure they do not overlap with the article title. (Page 1, Keywords, Line 31)
Comments 5: [Introduction: Tell us a little more about the ways of evaluating cytotoxicity in isolates from aquatic animals, especially Vibrio spp., and why cell lines from humans are used in this type of evaluation and not cell lines from aquatic animals.]
Response:Thank you very much for your careful review and question. V. parahaemolyticus is a pathogen that infects human intestinal epithelial cells. Therefore, we used cells of human origin rather than aquatic animal origin. Additionally, we have supplemented several ways of evaluating cytotoxicity of V. parahaemolyticus in the revised manuscript (Page 2, Introduction, Line69-78)
Comments6 :[Material and Methods: Lines 71-73: If the isolates used have been characterized in a previous study, please provide the appropriate citation. Otherwise, please explain how the samples were collected, how the bacteriological examination was performed and how they were identified at the species level.]
Response:We sincerely appreciate your careful reading! We have added details on sample collection, bacterial culture and bacterial identification in the revised version (Page 3-4, Material and Methods2.2, Line97-109)
Comments 7:[Material and Methods: Lines 78-79: add brand of culture media used.]
Response:Thanks for your careful checks. We feel sorry for our carelessness. In our resubmitted manuscript, we have added the brand name and item number of the media (Page3, Material and Methods2.2, Line103-104)
Comments 8:[Material and Methods: Line 83: describe in full the first time LB appears in the manuscript.]
Response:We are grateful for the suggestion. LB has been described in full as “Luria-Bertani liquid medium (LB) (Beijing Bio)” when it first appeared in the manuscript (Page 4, Material and Methods2.3, Line111-112)
Comments 9:[Material and Methods: Line 83-89: Was the experiment conducted in replicates? From Figure 1 it seems so, so I ask that you add the number of replicates tested.]
Response:Thank you for your careful reading and question. The experiment was repeated 3 times. We feel sorry for our carelessness. In our resubmitted manuscript, We have added the number of experiments (Page4, Material and Methods2.3, Line117)
Comments 10:[Material and Methods: Line 88: change "hole" to "well"; and was any analysis of colony forming unit counts performed on any of these ODs analyzed?]
Response:Thank you very much for your suggestion. We have changed the “hole” to “well”. In addition, the number of V. parahaemolyticus colonies was positively correlated with the ODs. However, we are sorry that we did not deeply analyze the quantitative relationship between ODs and colony forming units (Page4, Material and Methods2.3, Line116)
Comments 11:[Material and Methods: Line 98: Why didn't the authors perform the toxicity test with other concentrations (2 and 5%) of salt?]
Response:Thank you very much for your careful reading and question and sorry for the confusion. V. parahaemolyticus cultured at 0.5% or 1% sodium chloride solution has been reported to show higher cytotoxicity than at 3%, but the current study only compared two sets of salt concentrations, 0.5% versus 3% or 1% versus 3%. Combining the existing literature, in the present experiment, 0.5%, 1% and 3% were included in the same experiment for a comprehensive comparison.
Comments 12:[Material and Methods: Lines 118-120: This sentence should come at the beginning of the paragraph and not at the end.]
Response:Thank you for your valuable suggestion. The statement has been corrected (Page4, Material and Methods2.6, Line150-151)
Comments 13:[Material and Methods: Line 123: with 10 versions or version 10 of the program?]
Response:Thanks for your careful checks. We feel sorry for our carelessness. In our resubmitted manuscript, the version is revised (Page 5, Material and Methods2.7, Line158)
Comments 14:[Results: Line 130: delete "status"]
Response:Thank you very much for your valuable suggestion. We agree with you and have deleted "status" (Page 5, Results3.2, Line173)
Comments 15:[Results: Figure 1: improve the quality of the figure; subtitle suggestion: "Growth curves of freshwater and seawater group strains cultured in 0% (A), 0.5% (B), 1% (C), 2% (D), 3% (E), and 5% (F) sodium chloride LB liquid medium for 24 hours. Freshwater group strains (purple), seawater group strains (orange)."]
Response:Thank you for your comments. We have learned a lot from your comments. Based on your suggestions, we have upgraded the quality of the figure and modified the subtitle (Page 6, Results: Figure 3, Line179-182)
Comments 16:[Results: Line 158: change "Sodium" to "sodium". Make this change throughout the manuscript.]
Response: We sincerely appreciate your careful reading! We have changed the “Sodium” to “sodium” throughout the manuscript (Page7, Results: 3.3, Line195)
Comments 17:[Results: Figure 3: Why was only Caco-2 cell line chosen in image 3C?]
Response:Thank you for your careful reading and question. In reviewing the previous literature on V. parahaemolyticus cytotoxicity, the cells used include Caco-2 or Hela cell lines. In the experiments, cytotoxicity has been detected by LDH assay, and the CCK8 assay is mainly to further verify its consistency with the cytotoxicity results, so the cell viability assay was performed only with the Caco-2 cell line.
Comments 18:[Results:Table 1: move to after item 2.1; change “Regin” to “Region”]
Response:Thanks for your careful checks. We feel sorry for our carelessness. In our resubmitted manuscript, we've moved Table 1 after item 2.1 and revised “Regin” to “Region” (Page 3, Table 1, Line94)
Comments 19:[Discussion: Line 178: change "Haliphilism" to "Halophilism"; In addition, the word halophilism is repeated several times from the beginning of the discussion... would it be possible to change some of them for another synonym?]
Response: Thank you for your careful reading and question. We apologize for the poor language of our manuscript. We have corrected the “Haliphilism” into “Halophilism”. Besides, we have replaced the expression “Halophilism” with other synonyms, such as ‘salt dependence’, ‘salt adaptation’, etc (Page 8, Discussion, Line214,223,225)
Comments 20:[Discussion: Line 185: if OmpU and OmpN are referencing genes and not synthesized proteins, they should be left in italics.]
Response: Thanks for your careful checks. We are sorry for our carelessness. Based on your comments, we have made the corrections to mark “ompU and ompN” in italics (page8, Discussion, Line222)
Comments 21:[Discussion:Line 214: change “virulence” to “cytotoxicity”.]
Response: We sincerely appreciate your careful reading! We have changed “virulence” to “cytotoxicity” (page8, Discussion, Line253)
Comments 22:[Discussion: Line 221: What types of limitations? The authors mentioned that the growth of V. parahaemolyticus in different salt concentrations has already been evaluated... so, what is new about your study? The authors need to value the originality of the study carried out in the discussion.]
Response:We are grateful for the comment. We apologize for not expressing ourselves properly. We have modified this expression as follows: “The unique feature of this experiment lies in the successful screening of mutant strains through a series of salt concentration settings, and the cytotoxic difference of between freshwater and seawater strains at high and low salt concentrations were distinguished. Compared to previous experiments that only used standard strains. However, the strains used in the experiment only originated from Changzhou and were not expanded to other regions, so there are some limitations. Nevertheless, the results of this experiment provide some theoretical references for further research on V. parahaemolyticus. ” (Page 9, Discussion, Line 2579-266)
Comments 23:[Conclusions:Line 230: change "virulence" to "cytotoxicity"]
Response:Thanks for your careful checks. We revised the “cytotoxicity” instead of “virulence”. We will be happy to edit the text further based on your helpful comments (Page 9, Conclusions, Line 274)

Reviewer 2 Report
Comments and Suggestions for Authors
I provide comments and suggestions focusing on the M&M and leave the language improvement to the authors.
Heading 2.1:
Caco-2 (BeiNaBio, BNCC350769), HeLa (BeiNaBio, BNCC342189), and the V. parahaemolyticus standard strain (BeiNaBio, ATCC17802) were not shown in Table 1.
Table 1 should include the distinct characteristics or unique traits of each strain.
Heading 2.2:
Define the exact characteristics of ‘typical colonies.’
Heading 2.3:
Specify the type of growth curve, such as linear or curvilinear (e.g., logistic or Richards growth model). Please be specific.
Heading 2.5:
Indicators or measures of cytotoxicity, expressed as percentages, need to be transformed before any formal statistical analysis.
Heading 2.7:
Statistical analysis should be specific to each experiment, data type, or variable.
Include ANOVA tables as supplementary files to indicate the statistical significance of factors, such as using 2-way ANOVA to examine the effects of strain and sodium chloride levels.
Data were analysed and presented separately for fresh and saline water. It is essential to examine the effects of strain-by-environment interaction to understand whether the strain responds differently to each environment. The current analysis and results did not clearly show a comparison of halophilism and virulence or cytotoxicity between the two environments.
Indicate statistically significant differences between experimental groups or treatments in all tables and figures, and where appropriate, throughout the text.
Author Response
Response to Reviewer
Thank you very much for your insightful comments and valuable suggestions. The suggestions make our study more comprehensive and more rigorous. Based on the feedback provided, we have thoroughly revised the manuscript, and all authors have approved the changes. The main changes in the text are highlighted in red.
We have detailed the point-by-point response to the comments and suggestions below.
Comments 1:[Heading 2.1: Caco-2 (BeiNaBio, BNCC350769), HeLa (BeiNaBio, BNCC342189), and the V. parahaemolyticus standard strain (BeiNaBio, ATCC17802) were not shown in Table 1. Table 1 should include the distinct characteristics or unique traits of each strain.]
Response:Thank you for your careful reading and suggestions. As per your suggestion, we have added V. parahaemolyticus standard strain (BeiNaBio, ATCC17802) to Table I. Caco-2 and HeLa cells are not applicable to Table 1 (list of experimental strains), so they are listed in a separate paragraph. In addition, based on your suggestions, we have added the unique characteristics of the strains in Table 1 (page 3, Table1, Line94-95)
Comments 2:[Heading 2.2: Define the exact characteristics of ‘typical colonies.’]
Response:Thank you for your careful reading and question. Typical bacterial colonies refer to V. parahaemolyticus which showed round, smooth-surface green colonies on TCBS medium plate.
Comments 3:[Heading 2.3: Specify the type of growth curve, such as linear or curvilinear (e.g., logistic or Richards growth model). Please be specific.]
Response:We sincerely appreciate your suggestion. We apologize for not providing detailed information on the types of growth curves here. Therefore, we have made the following additions to the drawing of growth curves. “We measured the data once at each time point for two groups of strains and repeated it three times in parallel. The data from each time point were averaged as absorbance value. Fit a nonlinear regression curve with the cultivation time as the horizontal axis and the absorbance value of the culture as the vertical axis. Draw the halophilic growth curves of V. parahaemolyticus at different concentrations mentioned above.” (Page 4, Heading 2.3, Line117-122)
Comments 4: [Heading 2.5:Indicators or measures of cytotoxicity, expressed as percentages, need to be transformed before any formal statistical analysis.]
Response:Thank you very much for your careful reading and suggestions. For statistical analyses, we converted cytotoxicity to absolute values and multiplicities for between-group comparisons.
Comments 5:[Heading 2.7:Statistical analysis should be specific to each experiment, data type, or variable. Include ANOVA tables as supplementary files to indicate the statistical significance of factors, such as using 2-way ANOVA to examine the effects of strain and sodium chloride levels. Data were analyzed and presented separately for fresh and saline water. It is essential to examine the effects of strain-by-environment interaction to understand whether the strain responds differently to each environment. The current analysis and results did not clearly show a comparison of halophilism and virulence or cytotoxicity between the two environments. Indicate statistically significant differences between experimental groups or treatments in all tables and figures, and where appropriate, throughout the text.] Response:Thank you very much for your careful review and your suggestions, which have made our results more rigorous and better, and from which we have gained a great deal! We have supplemented Supplementary Material with a table of six sets of two independent samples t-test results for the two strains at six salt concentrations, as well as a table of one-way ANOVA for cytotoxicity, and the results of multiple comparisons. In the manuscript and in the Supplementary file, we have added the p-values for the experimental and control groups. In addition, we are sorry that we did not express ourselves clearly enough to give you a misunderstanding. We used the media containing different NaCl concentrations to simulate different environments. Analyzing the effects of different environments on growth adaptation and cytotoxicity of strains by culturing them in different environments to analyze the growth and cytotoxicity of the same type of strains at different environments. In addition, wild-type strains responded differently in different environments, which was used as a control group to analyze the effects of environmental changes on, among other things, the growth adaptation of freshwater strains. We will be happy to edit the text further based on your helpful comments.

Round 2
Reviewer 2 Report
Comments and Suggestions for Authors
The revisions are satisfactory.